# Evaluation of the Effectiveness and Use of Anti-Methicillin-Resistant *Staphylococcus aureus* Agents for Aspiration Pneumonia in Older Patients Using a Nationwide Japanese Administrative Database

**DOI:** 10.3390/microorganisms11081905

**Published:** 2023-07-27

**Authors:** Satoru Koga, Takahiro Takazono, Takashi Kido, Keiji Muramatsu, Kei Tokutsu, Takatomo Tokito, Daisuke Okuno, Yuya Ito, Hirokazu Yura, Kazuaki Takeda, Naoki Iwanaga, Hiroshi Ishimoto, Noriho Sakamoto, Kazuhiro Yatera, Koichi Izumikawa, Katsunori Yanagihara, Yoshihisa Fujino, Kiyohide Fushimi, Shinya Matsuda, Hiroshi Mukae

**Affiliations:** 1Department of Respiratory Medicine, Nagasaki University Hospital, Nagasaki 852-8501, Japan; 2Department of Infectious Diseases, Nagasaki University Graduate School of Biomedical Sciences, Nagasaki 852-8523, Japan; 3Department of Preventive Medicine and Community Health, University of Occupational and Environmental Health, Japan, Kitakyusyu 807-8555, Japan; 4Department of Respiratory Medicine, University of Occupational and Environmental Health, Japan, Kitakyusyu 807-8555, Japan; 5Department of Laboratory Medicine, Nagasaki University Graduate School of Biomedical Sciences, Nagasaki 852-8523, Japan; 6Department of Environmental Epidemiology, Institute of Industrial Ecological Science, University of Occupational and Environmental Health, Japan, Kitakyusyu 807-8555, Japan; 7Department of Health Policy and Informatics, Graduate School of Medical and Dental Sciences, Tokyo Medical and Dental University, Tokyo 113-8510, Japan

**Keywords:** anti-MRSA agents, aspiration pneumonia, older patients, hospitalization

## Abstract

Studies indicated potential harm from empirical broad-spectrum therapy. A recent study of hospitalizations for community-acquired pneumonia suggested that empirical anti-methicillin-resistant *Staphylococcus aureus* (MRSA) therapy was associated with an increased risk of death and other complications. However, limited evidence supports empirical anti-MRSA therapy for older patients with aspiration pneumonia. In a nationwide Japanese database, patients aged ≥65 years on admission with aspiration pneumonia were analyzed. Patients were divided based on presence of respiratory failure and further sub-categorized based on their condition within 3 days of hospital admission, either receiving a combination of anti-MRSA agents and other antibiotics, or not using MRSA agents. An inverse probability weighting method with estimated propensity scores was used. Out of 81,306 eligible patients, 55,098 had respiratory failure, and 26,208 did not. In the group with and without respiratory failure, 0.93% and 0.42% of the patients, respectively, received anti-MRSA agents. In patients with respiratory failure, in-hospital mortality (31.38% vs. 19.03%, *p* < 0.001), 30-day mortality, and 90-day mortality were significantly higher, and oxygen administration length was significantly longer in the anti-MRSA agent combination group. Anti-MRSA agent combination use did not improve the outcomes in older patients with aspiration pneumonia and respiratory failure, and should be carefully and comprehensively considered.

## 1. Introduction

Pneumonia is a major cause of death in aging countries such as Japan and the United Kingdom, and its incidence is increasing with an aging population [1,2]. In particular, the ratio of aspiration pneumonia to total pneumonia cases increases with age, and it has been reported that approximately 80% of patients with pneumonia aged >70 years are diagnosed with aspiration pneumonia [3]. A recent study reported that 89% of patients whose deaths were related to aspiration pneumonia were over the age of 65 [4].

Since aspiration pneumonia in older patients is frequently lethal, initiation of empirical broad-spectrum therapy is often considered. However, studies have suggested that empirical broad-spectrum therapy for community-acquired pneumonia may be harmful [5]. Furthermore, the increase in antimicrobial resistance owing to the abuse of broad-spectrum antibiotics has become a global issue. Nevertheless, American Thoracic Society (ATS)/Infectious Diseases Society of America (IDSA) and Japanese guidelines recommend the empirical use of anti-methicillin-resistant *Staphylococcus aureus* (MRSA) agents, considering the severity, history of MRSA isolation, and recent parenteral antibiotic therapy [6,7]. However, MRSA isolated from patients with pneumonia may be due to colonization, and may not be the causative agent of pneumonia in most MRSA culture-positive cases [8]. In addition, a recent retrospective cohort study of all hospitalizations for community-acquired pneumonia also suggested that empirical anti-MRSA therapy plus standard therapy was significantly associated with an increased adjusted risk of death and other complications [5]. 

MRSA or other antimicrobial-resistant bacteria are more frequently detected in older patients with aspiration pneumonia [9,10] than in community-acquired pneumonia. Evidence of using empirical anti-MRSA therapy plus standard therapy for older patients with aspiration pneumonia is limited. However, it may be difficult to conduct large, prospective, and randomized control trials in the real-world setting for older patients with aspiration pneumonia due to the difficulty of triage, costs, and/or ethical issues. Therefore, we investigated the data from clinical practice on the use of concomitant anti-MRSA agents and their effectiveness for aspiration pneumonia in older patients using the Diagnostic Procedure Combination (DPC), a nationwide Japanese administrative database. 

## 2. Materials and Methods

### 2.1. Database

We used the DPC database to investigate the use and effectiveness of concomitant anti-MRSA agents in patients aged ≥65 years with aspiration pneumonia. The DPC research group, which receives funding from the Japanese Ministry of Health, Labor, and Welfare, extracted all data from the DPC database. The DPC is a Japanese case-mix patient classification system launched in 2002 for payment management and the modernization of the healthcare system. The database contains the following details: patient age, sex, diagnosis, comorbidities at admission and during hospitalization (coded following the International Classification of Diseases, 10th revision; ICD-10), state of consciousness according to the Japan Coma Scale, medical procedures, medications, intensive-care unit (ICU) admission, interventional procedures (including medical ventilation, and heart-lung medicine), length of hospital stay, and discharge status (including in-hospital deaths) [11,12]. 

### 2.2. Patient Consent Statement

The requirement for informed consent was waived because any personally identifiable information was removed from the extracted data and because of the study’s retrospective nature. This study was approved by the Ethics Committee of Medical Research, University of Occupational and Environmental Health, Japan (registration number R04–045), and conducted according to the guidelines of the Declaration of Helsinki. 

### 2.3. Patient Selection

The study population consisted of patients aged ≥ 65 years on admission; based on the study, of the patients whose death was aspiration pneumonia associated, 89% were age 65 or older [4], who were admitted to the hospital due to aspiration pneumonia (IDC-10 code J690). 

Data of patients admitted between January and December 2018 who met the criteria were extracted from the DPC database. 

The patients were divided into two groups according to their condition within 3 days of hospital admission: with respiratory failure and without respiratory failure, based on their oxygen therapy requirement. In this study, patients who needed oxygen therapy were defined as those with respiratory failure and those who did not require oxygen therapy were defined as those without respiratory failure. They were further divided into the anti-MRSA agent combination use group (anti-MRSA agent group) and the non-anti-MRSA agent group. The anti-MRSA agent group included patients who received anti-MRSA agents (vancomycin, teicoplanin, arbekacin, and linezolid: these antimicrobials are approved in Japan as anti-MRSA agents and CLSI breakpoints are used) in combination with other antimicrobials within 3 days of hospital admission and non-anti-MRSA agent group included patients who received antimicrobials other than anti-MRSA agents within 3 days of admission. Patients treated with anti-MRSA agents only and those who did not receive any antibiotics were excluded from this study. Patients with missing data were also excluded from this study.

### 2.4. Variables

The following factors were used as variables at admission: age (at hospital admission), sex, emergency admission, emergency transport, respiratory medicine (the main department), advanced treatment hospital (hospitals required to provide advance medical care, most of which are university hospitals), dementia (extracted from information about older people in DPC entry field, not from registered ICD-10 diseases), coma (Japan coma scale score at admission ≥100 which is approximately equal to Glasgow Coma Scale (GCS) E1V1M5), chronic obstructive pulmonary disease (COPD), asthma, diabetes, chronic kidney disease, pulmonary aspergillosis, non-tuberculosis mycobacteria (NTM), interstitial pneumonia (IP), Parkinson’s syndrome, sequelae of cerebrovascular disease, abscess of lung and mediastinum, pyothorax (including empyema), achalasia, malignant neoplasms, malignant neoplasms in the oral cavity, malignant neoplasm of the esophagus, additional reimbursement for infection control team (ICT), and additional reimbursement for antimicrobial stewardship team (AST). The following variables were assessed within 3 days of hospital admission: maintenance dialysis, emergency dialysis, surgery, ICU admission, high-flow nasal cannula (HFNC), non-invasive positive pressure ventilation (NPPV), invasive positive pressure ventilation (IPPV), total parenteral nutrition (TPN), nasal tube feeding, gastrostomy nutrition, dysphagia rehabilitation, and medication use (including dopamine, noradrenaline, steroids, insulin, albumin, immunoglobulin, anti-influenza virus drug, intravenous administration of penicillin, macrolides, cephalosporins, carbapenems, clindamycin, minocycline, metronidazole, and intravenous and oral administration of new quinolones). Additional reimbursements for ICT and AST are provided to hospitals that continuously conduct the appropriate activities required for infection control and antimicrobial stewardship.

### 2.5. Outcomes

The primary endpoint was mortality (in-hospital, 30-day, and 90-day mortalities). The secondary endpoints were the length of hospital stay, duration of oxygen administration, and ICU admission. For patients without respiratory failure, no outcome comparisons were performed for the duration of oxygen administration.

### 2.6. Statistical Analysis

The propensity score was calculated using a logistic model with baseline variables that could affect the administration of anti-MRSA agents and outcomes, including all variables described above. 

The C-statistic (area under the operating characteristic curve) was employed to evaluate the goodness of the fit. We then adjusted for cofounders and evaluated the outcomes between the anti-MRSA agent and non-anti-MRSA agent groups based on pneumonia severity (with or without respiratory failure group) using the inverse probability weighting (IPW) method [11,13,14]. The covariates before adjustment were evaluated using the chi-squared test for categorical variables and the unpaired *t*-test for continuous variables. These analyses were conducted using STATA/MP17 (Stata Corp, College Station, TX, USA). Results with *p* < 0.05 were considered statistically significant in all tests.

## 3. Results

A flowchart of patient selection for aspiration pneumonia is illustrated in Figure 1. The data of 89,595 patients aged ≥65 years with aspiration pneumonia were extracted from the DPC database, as described in the methods section. Of these, 8289 patients not receiving antibiotics other than anti-MRSA agents within 3 days of hospital admission or without information on antimicrobial agents were excluded, while 81,306 were eligible for this study. Of the 81,306 eligible patients, 55,098 (67.72%) were administered oxygen therapy within 3 days of hospital admission. In the with respiratory failure and without respiratory failure groups, anti-MRSA agents were used in 515 (0.93%) and 111 (0.42%) patients, respectively.

In an analysis of all 81,306 cases enrolled in this study, penicillin, third-generation cephalosporins, and carbapenems were used in 75.18%, 19.38%, and 5.12% of cases, respectively. The numbers of concomitant antibiotics in patients who received multiple antibiotics were two (91.02%), three (8.42%), and four or more types of antibiotics (0.57%). The frequencies of concomitant antibiotics with anti-MRSA agents were penicillin (59.90%), third-generation cephalosporins (14.70%), and carbapenems (35.14%). The numbers of concomitant antibiotics in patients using anti-MRSA agents were two (79.55%), three (19.01%), and more than four (1.44%), indicating that the anti-MRSA agent group was administered multiple drugs more than the non-anti-MRSA agent group.

Subsequently, we used the IPW method with estimated propensity scores; the C-statistic of the propensity score was 0.795 for patients with respiratory failure and 0.8144 for those without respiratory failure. The baseline characteristics of the patients before and after adjusting for confounders are presented in the Appendix A.

In patients with respiratory failure, before adjustment, the baseline variables of age, sex, emergency transport, advanced treatment hospital, dementia, coma, COPD, asthma, maintenance dialysis, emergency dialysis, malignant neoplasms, malignant neoplasms in the oral cavity, additional reimbursement for ICT, additional reimbursement for AST, surgery, IPPV, HFNC, NPPV, ICU admission, dopamine, noradrenaline, steroids, insulin, albumin, immunoglobulin, TPN, nasal tube feeding, gastrostomy nutrition, penicillin, macrolides, third-generation cephalosporins, and carbapenems differed significantly between the anti-MRSA agent combination and non-anti-MRSA agent groups. After adjusting for confounders, the patient baseline characteristics in both groups were similar across these variables, except for dementia, NTM, pyothorax, achalasia, malignant neoplasm in the oral cavity, malignant neoplasm of the esophagus, and additional reimbursement for ICT and IPPV, which remained significantly different.

In contrast, in patients without respiratory failure, the baseline variables of age, sex, emergency transport, advanced treatment hospital, neurological dysfunction, maintenance dialysis, pyothorax, malignant neoplasms in the oral cavity, additional reimbursement for AST, surgery, ICU admission, steroid, insulin, albumin, immunoglobulin, anti-influenza virus drug, TPN, nasal tube feeding, penicillin, macrolides, third-generation cephalosporins, fourth-generation cephalosporins, and carbapenems differed significantly between the anti-MRSA agent combination and the non-anti-MRSA agent groups. After adjusting for confounders, the patient baseline characteristics in both groups were similar across these variables, except for dementia, diabetes, NTM, IP, maintenance dialysis, pyothorax, malignant neoplasms, and additional reimbursement for ICT, which remained significantly different.

The results of the primary and secondary endpoints of patients treated with an anti-MRSA agent combination and non-anti-MRSA agents, as assessed by the IPW method with the estimated propensity score, are presented in Table 1. 

In patients with respiratory failure, in-hospital mortality and 30-day and 90-day mortality rates were significantly higher, and the length of oxygen administration was significantly longer in the anti-MRSA agent group. The lengths of hospital stay and ICU admission were not significantly different between both groups.

In patients without respiratory failure, the outcomes were not significantly different between groups.

## 4. Discussion

In this study, we investigated clinical practice evidence of the use and effectiveness of anti-MRSA agent combined therapy in 81,306 patients with aspiration pneumonia with or without respiratory failure using the nationwide Japanese administrative database. The rate of combination use of anti-MRSA agents within 3 days of hospital admission for aspiration pneumonia was 0.93% and 0.42% in patients with and without respiratory failure, respectively. Patients who were treated with anti-MRSA agents were treated with penicillin (59.90%), third-generation cephalosporins (14.70%), and carbapenems (35.14%); the number of concomitant antibiotics in patients treated with anti-MRSA agents was two (79.55%), three (19.01%), and more than four (1.44%), indicating that the anti-MRSA agents group was administered broad-spectrum and multiple antibiotics. This study revealed that in-hospital mortality, 30-day mortality, and 90-day mortality rates were significantly higher, and the length of oxygen administration was significantly longer in the anti-MRSA agent group in patients with respiratory failure.

In a retrospective multicenter cohort study of patients with community-acquired pneumonia, empirical anti-MRSA therapy plus standard therapy was associated with an increased risk of death, kidney injury, secondary *Clostridioides difficile* infections, vancomycin-resistant *Enterococcus* spp. infections, and gram-negative rod infections [5]. The use of anti-MRSA antibiotics can result in adverse drug events. For example, vancomycin has been associated with nephrotoxicity in numerous studies, with rates ranging from 5 to 43% [15,16], and linezolid causes hematological toxicity, mostly thrombocytopenia, which occurs in 28.9–78.6% and 10.5–42.9% of patients with and without decreased renal function (DRF), respectively [17]. And as low Creatinine Clearance (C-Cre) is a risk for developing linezolid-induced thrombocytopenia [18], it may be more likely to occur in Japanese individuals with lower C-Cre due to lower body weight. As patients with aspiration pneumonia are an older population with more comorbidities, the adverse events described above might negatively impact their prognosis.

Several previous reports have questioned the empirical use of broad-spectrum antimicrobial agents, including anti-MRSA agents, in community-acquired pneumonia [5,19]. Our findings provided similar results even in patients with aspiration pneumonia, whose population is likely to be at risk of bacterial resistance, as it is more common in healthcare-associated pneumonia than in community-acquired pneumonia [20]. It is important to note that this study did not focus on whether MRSA should be treated when isolated, but only evaluated the benefit of empirical therapy for MRSA in the initial treatment of aspiration pneumonia.

The only large clinical study that has analyzed anti-MRSA agents used for pneumonia is a report using the Veterans Administration dataset for patients with community-acquired pneumonia (38%) [5]. Surprisingly, the results of our study revealed a much lower rate of anti-MRSA agent use (0.93% and 0.42%, with and without respiratory failure, respectively) than that reported in a previous study [5]. Two factors may influence the difference in the rate of anti-MRSA agent use. First, while the majority of CA-MRSA in Europe and the United States are PVL-positive, causing serious infections [21,22], the PVL-positive rate of CA-MRSA in Japan is lower than in Western countries [23]. Second, the guidelines in Japan on pneumonia management in adults might have an impact. These guidelines recommend that decisions regarding therapy for patients with aspiration pneumonia should be based on respect for the individual’s wishes and quality of life before considering aggressive antimicrobial therapy [6].

Almost all variables were adjusted; however, some revealed significant differences between the two groups after weighting. Notwithstanding, these variables may be biased since they have a very low incidence. Therefore, these results were not considered meaningful. The variables for which significant differences remained that could worsen the prognosis of aspiration pneumonia or suggest severe pneumonia (dementia, NTM, pyothorax, achalasia, malignant neoplasms in the oral cavity, malignant neoplasms of the esophagus, and IPPV in patients with respiratory failure and diabetes, NTM, IP, maintenance dialysis, pyothorax, and malignant neoplasms in patients without respiratory failure) were fewer in the anti-MRSA agent combination group. In addition, as the variables used to evaluate pneumonia severity (age, coma, IPPV, NPPV use, vasopressors (dopamine, noradrenaline) use, and ICU admission) were adjusted for PS, we consider that the severity of pneumonia did not significantly affect the results of this study.

This study had several limitations similar to previous studies that used the DPC database [11,24,25,26]. First, this was an observational retrospective study without randomization. Adjusting the baseline characteristic differences using the IPW method with the estimated propensity score reduced the effect of this limitation; however, there might have been confounders that we were unable to evaluate using the DPC database that could have influenced the outcome. Such confounders include vital signs, laboratory findings, radiological findings, and the amount of oxygen administered. Second, this study did not consider the type, dose, and duration of anti-MRSA agents. Third, the influence of bacteriological test results (including the results of culture testing and susceptibility testing), recent antimicrobial exposure history before admission, and previous history of MRSA infections which are useful for predicting MRSA-caused pneumonia [27,28] were not considered because the DPC data we extracted does not include this information. 

## 5. Conclusions

Evidence of the effects of empirical anti-MRSA therapy plus standard therapy for older patients with aspiration pneumonia is limited. Our results revealed that the rate of anti-MRSA agent use within 3 days of hospital admission for aspiration pneumonia in older patients may be much lower than that reported for community-acquired pneumonia in the United States. Our study also revealed that anti-MRSA agent use did not improve outcomes in patients with respiratory failure, consistent with previous studies on the empirical use of broad-spectrum antimicrobial agents, including anti-MRSA agents, for community-acquired pneumonia. This study only applies “in general” because of some limitations due to the nature of the DPC database.

Further prospective studies on aspiration pneumonia in older patients are required to confirm our results.

## Figures and Tables

**Figure 1 microorganisms-11-01905-f001:**
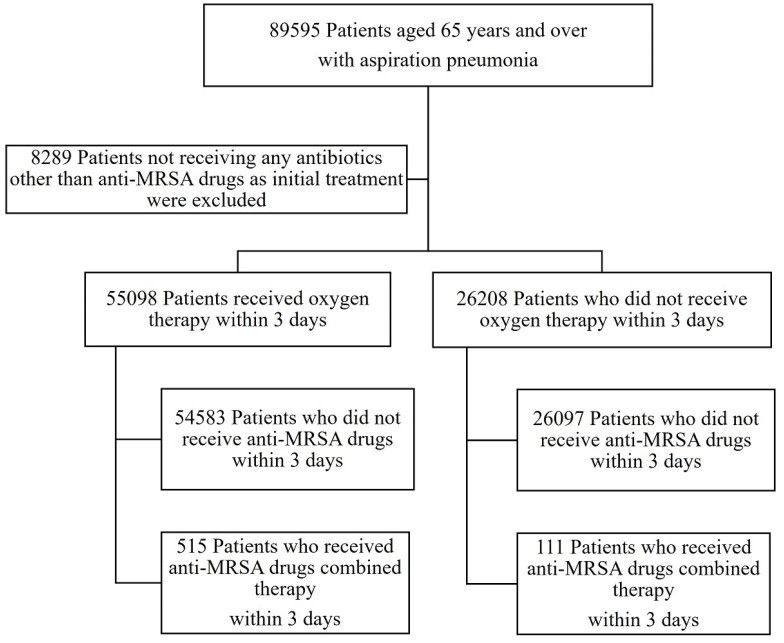
Flow chart of patient selection. Between January and December 2018, 89,595 patients were hospitalized for aspiration pneumonia. Of these, 8289 patients were excluded from the study because they received no antimicrobial therapy or received anti-Methicillin-resistant *Staphylococcus aureus* (MRSA) agents only, within 3 days of admission. Of the remaining patients, 55,098 were on oxygen within 3 days of admission and 26,208 were not. In the respiratory failure and moderate groups, 515 (0.93%) and 111 (0.42%) patients, respectively, received anti-MRSA agent combined therapy within 3 days of admission.

**Table 1 microorganisms-11-01905-t001:** Results of the primary and secondary endpoints of patients treated with or without anti-MRSA agents; with respiratory failure group (a) and without respiratory failure group (b).

**(a)**			
**Outcomes**	**Anti-MRSA Agents Group**	**Non-Anti-MRSA Agents**	***p*-Value**
In-hospital mortality (%)	31.38	19.96	<0.001
30-day mortality (%)	22.52	14.11	<0.001
90-day mortality (%)	29.97	19.03	<0.001
Hospital stay, mean days (SE)	29.66 (1.80)	27.74 (0.289)	0.289
Oxygen administration, mean days (SE)	11.41 (0.81)	9.69 (0.06)	0.035
ICU admission, mean days (SE)	0.06 (0.01)	0.07 (0.003)	0.13
**(b)**			
**Outcomes**	**Anti-MRSA Agents Group**	**Non-Anti-MRSA Agents**	***p*-Value**
In hospital mortality (%)	14.70	9.07	0.315
30-day mortality (%)	12.62	4.3	0.129
90-day mortality (%)	14.70	8.10	0.238
Hospital stay, mean days (SE)	27.60 (2.08)	26.19 (0.498)	0.498
ICU admission, mean days (SE)	0.79 (0.004)	1.89 (0.002)	0.420

ICU, intensive care unit; MRSA, methicillin-resistant Staphylococcus aureus; SE, standard error.

## Data Availability

Data supporting reported results will be provided upon request.

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
