# Peer review of "Evaluation of the Effectiveness and Use of Anti-Methicillin-Resistant Staphylococcus aureus Agents for Aspiration Pneumonia in Older Patients Using a Nationwide Japanese Administrative Database"

_microorganisms, 2023, doi:10.3390/microorganisms11081905_

Round 1
Reviewer 1 Report
Major points
1. The higher mortality rates and longer oxygen administration observed in the anti-MRSA agent group in the patients with respiratory failure can be attributed to the disease severity of patients, which is consistent with the results that the anti-MRSA agents were more frequently administered with carbapenems and multiple antibiotics.
The authors should mention in the Discussion about the impact of the differences in the severity between patients who received anti-MRSA agents and those who did not, on the study results.
2. The authors used the IPW method with the estimated propensity score, but some of the variables after adjustment were not consistent with those before adjustment.
Table 1: Why was the rate of IPPV use significantly higher in the anti-MRSA agent’s combination group before adjustment, but reversed (significantly lower) after adjustment, as well as Dementia and NTM in the patients with respiratory failure? Are these statistically correct?
3. Table 1, 2: In this study, many variables were evaluated and these data are very important. The authors found that in-hospital mortality, 30-day mortality, and 90-day mortality rates were significantly higher in the anti-MRSA agent group in patients with respiratory failure.
Is the use of anti-MRSA agents an independent factor increasing mortality? Are there other important factors associated with increasing mortality? It is better to conduct multivariable analyses to identify the independent factors associated with in-hospital, 30-day, and 90-day mortality.
3. Line 262: The incidence of linezolid-induced thrombocytopenia appears to be higher than reported. The authors should cite recent studies and revise the occurrence rate, taking into account of older age and lower body weight in the Japanese population.
4. Lines 278–282: I think the author misunderstands the “PVL-positive CA-MRSA” as “CA-MRSA”.
CA-MRSA harbors SCCmec types IV, V, VI, VII, and VIII are dominant even in Japan recently. Therefore, these sentences might mislead the readers. The authors should revise and correct these sentences.
Minor points
5. Title: Given the difference in meaning between “efficacy” and “effectiveness”, it seems that “efficacy” should be changed to “effectiveness”.
Author Response
Response to Reviewer 1 Comments
- The higher mortality rates and longer oxygen administration observed in the anti-MRSA agent group in the patients with respiratory failure can be attributed to the disease severity of patients, which is consistent with the results that the anti-MRSA agents were more frequently administered with carbapenems and multiple antibiotics.
The authors should mention in the Discussion about the impact of the differences in the severity between patients who received anti-MRSA agents and those who did not, on the study results.
Response 1:
Thank you for your valuable comments. As you pointed out, anti-MRSA agent combination group before adjustment by PS had more severe pneumonia in terms of age, coma, IPPV, NPPV use, vasopressors (dopamine, noradrenaline) use, and ICU admission which are used for evaluating the severity of pneumonia. However, these covariates were adjusted for PS. Therefore, the anti-MRSA combination group is not considered as having a more severe condition after adjustment.
We added the sentences in line 289-292 of Discussions as below: In addition, as the variables used to evaluate pneumonia severity (age, coma, IPPV, NPPV use, vasopressors (dopamine, noradrenaline) use, and ICU admission) were adjusted, we consider that the severity of pneumonia did not significantly affect the results of this study.
- The authors used the IPW method with the estimated propensity score, but some of the variables after adjustment were not consistent with those before adjustment.
Table 1: Why was the rate of IPPV use significantly higher in the anti-MRSA agent’s combination group before adjustment, but reversed (significantly lower) after adjustment, as well as Dementia and NTM in the patients with respiratory failure? Are these statistically correct?
Response 2:
Some of the covariates remained significantly different after adjustment, however, we consider these differences for the covariates you pointed out to be minor as the following reasons:
1. Dementia and IPPV use: we consider the probability of those covariates after adjustment to be appropriately adjusted because they are closer than before adjustment.
2. NTM: the probability is extremely low close to 0 both before and after adjustment.
The IPTW method is well established one and this study was conducted in consultation with a statistical expert. Therefore, we believe that the results are statistically correct, even if the adjusted results are reversed.
- Table 1, 2: In this study, many variables were evaluated and these data are very important. The authors found that in-hospital mortality, 30-day mortality, and 90-day mortality rates were significantly higher in the anti-MRSA agent group in patients with respiratory failure.
Is the use of anti-MRSA agents an independent factor increasing mortality? Are there other important factors associated with increasing mortality? It is better to conduct multivariable analyses to identify the independent factors associated with in-hospital, 30-day, and 90-day mortality.
Response 3:
Thank you for your valuable feedback and thoughtful considerations. We appreciate your attention to the data and the important issue of mortality in our study.
Indeed, this study aims to investigate the association between antibiotic administration and mortality in patients with respiratory failure. We used the propensity score method instead of multivariable analysis to adjust for confounding factors, both of which are commonly used approaches for adjustment.
Based on the results, we can assert that the use of anti-MRSA agents is associated with mortality, at least independently of the factors used in the propensity score.
We understand your concern regarding individual factors in multivariable analysis and their potential for causing misinterpretations in terms of causal estimation. In this study, we considered multiple factors as confounding variables in assessing the association between antibiotics and mortality. However, the associations of these factors with mortality do not directly explain the relationship between antibiotics and mortality.
Thus, we agree that mentioning the associations of other factors with mortality in this study would be better avoided. We have carefully discussed and considered this issue with a statistical expert to ensure the accuracy of our response.
- Line 262: The incidence of linezolid-induced thrombocytopenia appears to be higher than reported. The authors should cite recent studies and revise the occurrence rate, taking into account of older age and lower body weight in the Japanese population.
Response 4:
Thank you for your very valuable suggestions. In response to your suggestion, we have changed the sited references to more up-to-date ones and revised the discussions in line 254-258 as follows:
Linezolid causes hematological toxicity, mostly thrombocytopenia, which occurs in patients with and without decreased renal function (DRF) in 28.9-78.6% and 10.5-42.9% respectively [1]. And as the low C-Cre is a risk for developing linezolid-induced thrombocytopenia [2], it may be more likely to occur in Japanese with lower C-Cre due to lower body weight.
Cited reference:
1) Liu X, Aoki M, Osa S, et al. Safety of linezolid in patients with decreased renal function and trough monitoring: a systematic review and meta-analysis. BMC Pharmacology and Toxicology 2022; 23(1): 89.
2) Matsumoto K, Takeda Y, Takeshita A, et al. Renal function as a predictor of linezolid-induced thrombocytopenia. Interna-tional journal of antimicrobial agents 2009; 33(1): 98-9.
- Lines 278–282: I think the author misunderstands the “PVL-positive CA-MRSA” as “CA-MRSA”.
CA-MRSA harbors SCC mec types IV, V, VI, VII, and VIII are dominant even in Japan recently. Therefore, these sentences might mislead the readers. The authors should revise and correct these sentences.
Response 5:
Thank you so much for pointing out the misleading representations regarding CA-MRSA.
We revised the discussions and added the references in line 274-276 as below:
First, while the majority of CA-MRSA in Europe and the USA are PVL-positive causing serious infections [1.2], the PVL-positive rate of CA-MRSA is lower in Japan than in Western countries [3].
Cited reference:
1) Liu X, Aoki M, Osa S, et al. Safety of linezolid in patients with decreased renal function and trough monitoring: a systematic review and meta-analysis. BMC Pharmacology and Toxicology 2022; 23(1): 89.
2) Matsumoto K, Takeda Y, Takeshita A, et al. Renal function as a predictor of linezolid-induced thrombocytopenia. International journal of antimicrobial agents 2009; 33(1): 98-9.
3) David MZ, Daum RS. Community-associated methicillin-resistant Staphylococcus aureus: epidemiology and clinical consequences of an emerging epidemic. Clinical microbiology reviews 2010; 23(3): 616-87.
- Title: Given the difference in meaning between “efficacy” and “effectiveness”, it seems that “efficacy” should be changed to “effectiveness”.
Response 6:
Thank you for your valuable suggestion. I completely agree with your opinion. We revised our manuscript title as below: Evaluation of the effectiveness and use of anti-methicillin-resistant Staphylococcus aureus agents for aspiration pneumonia in older patients using a nationwide Japanese administrative database.

Reviewer 2 Report
Koga et al. have presented a database-based retrospective assessment on the inclusion of anti-MRSA antibiotics in the treatment of aspiration pneumonia in Japan. The authors have openly admitted the study’s limitations at the end of the discussion, mainly resulting from an insufficient depth and discriminatory power of the data available within the database. Before publishing is considered, I have a few recommendations on how the manuscript might be further improved.
1.) The authors have already mentioned that stratification by antimicrobial substance is unfeasible due to database limitations. However, they should consider stratification by patient groups. Are there any subgroups with higher MRSA rates, so beneficial effects of adding MRSA-active agents might be observed just within such sub-groups? In case that such a stratification is unfeasible as well, the authors should at least explain why in the limitations paragraph of their discussion.
2.) Methods chapter: When mentioning the assessed anti-MRSA agents (vancomycin, teicoplanin, arbekacin, linezolid), the authors should explain which interpretation standard (CLSI?) is applied for the interpretation of susceptibility testing results in their country.
3.) Results: Please provide numbers and denominators (n/n) instead of mere percentage values to make the interpretation of the results easier for the reader. Percentages based on small denominators might be less reliable.
4.) Tables: The tables are very comprehensive but difficult to reader. The authors should consider presenting them as appendix tables at the end of the manuscript and summarizing the most important results in smaller tables or reader-friendly diagrams.
5.) Conclusion, last sentence: The authors should make clear that – due to the limitations as stated above – this sentence only applies “in general”. So far, their study does not exclude potential benefits for certain sub-groups with higher MRSA rates.
Author Response
Response to Reviewer 2 Comments
1.The authors have already mentioned that stratification by antimicrobial substance is unfeasible due to database limitations. However, they should consider stratification by patient groups. Are there any subgroups with higher MRSA rates, so beneficial effects of adding MRSA-active agents might be observed just within such sub-groups? In case that such a stratification is unfeasible as well, the authors should at least explain why in the limitations paragraph of their discussion.
Response 1:
The most consistently strong individual risk factors for respiratory infection with MRSA are prior isolation of these organisms, especially from the respiratory tract, and/or recent hospitalization and exposure to parenteral antibiotics [1, 2]. As you pointed out, we believe that stratifying patients by risk factors for MRSA infection would make this study more valuable.
The DPC database basically includes the data of patients’ diagnosis, medical procedure, and medications used during the period of hospitalization. However, unfortunately, it does not include the outpatient data; hence, it is difficult to link the data in this study with the previous hospitalization data. We revised the sentence of Discussions in line 300-305 as below: Third, the influence of bacteriological test results (including the results of culture testing and susceptibility testing), recent antimicrobial exposure history before admission, and previous history of MRSA infections which are useful for predicting MRSA caused pneumonia [1.2] were not considered because the DPC data we extracted does not include this information.
Cited reference:
1) Webb BJ, Dascomb K, Stenehjem E, Dean N. Predicting risk of drug-resistant organisms in pneumonia: moving beyond the HCAP model. Respiratory medicine 2015; 109(1): 1-10.
2) Aliberti S, Reyes LF, Faverio P, et al. Global initiative for methicillin-resistant Staphylococcus aureus pneumonia (GLIMP): an international, observational cohort study. The Lancet Infectious diseases 2016; 16(12): 1364-76.
- Methods chapter: When mentioning the assessed anti-MRSA agents (vancomycin, teicoplanin, arbekacin, linezolid), the authors should explain which interpretation standard (CLSI?) is applied for the interpretation of susceptibility testing results in their country.
Response 2:
Thank you for your valuable suggestion. As you pointed out, we agree that our descritption was insufficient explanation as teicoplanin and arbekacin are not approved as anti-MRSA drugs outside Japan. We added the sentences in line 108-109 as below: these antimicrobials are approved in Japan as anti-MRSA agents and CLSI breakpoints are used for the evaluation of drug sensitivity test.
- Results: Please provide numbers and denominators (n/n) instead of mere percentage values to make the interpretation of the results easier for the reader. Percentages based on small denominators might be less reliable.
Response 3:
Thank you for your feedback regarding the presentation of results. We appreciate your concern about the use of Propensity Score and the IPTW method in our study. We understand that providing numbers and denominators (n/n) alongside percentage values may improve the interpretability of the results. However, it is important to note that the IPTW method is used to balance the distribution of covariates between treatment groups and create a pseudo-population. This technique aims to estimate the treatment effect without bias due to confounding factors.
In the context of the IPTW method, presenting the adjusted numbers may not be appropriate as they are not directly representative of the actual population size. The weights applied to each individual can lead to non-integer values, and this might confuse readers if presented in the results table. The primary purpose of utilizing the IPTW method is to derive unbiased estimates of treatment effects, and the focus is typically on presenting the effect size and its statistical significance, rather than raw numbers.
In addition, we need to apologize for one point. In the process of revision, we realized that we had an inadvertent mistake in copying the data of unadjusted items of Appendix table (b) in the first submission. However, this mistake does not affect the result of this study. We have corrected these mistakes in this revised version.
- Tables: The tables are very comprehensive but difficult to reader. The authors should consider presenting them as appendix tables at the end of the manuscript and summarizing the most important results in smaller tables or reader-friendly diagrams.
Response 4:
Thank you for your valuable suggestion. We agree with your opinion as our Table1 is very comprehensive. As we consider the important content about Table1 is described in the text, we transferred the Table1 to Appendix table. Accordingly, Table2 was renamed as Table1.
- Conclusion, last sentence: The authors should make clear that – due to the limitations as stated above – this sentence only applies “in general”. So far, their study does not exclude potential benefits for certain sub-groups with higher MRSA rates.
Response 5:
Thank you for your valuable comment. We completely agree with your suggestion. We added the sentence in line 314-315 of the conclusions as below:
This study only applies “in general” because of some limitations due to the nature of the DPC database.

Round 2
Reviewer 1 Report
Thank you for your time and careful responses.